# Fabrication, Structural and Biological Characterization of Zinc-Containing Bioactive Glasses and Their Use in Membranes for Guided Bone Regeneration

**DOI:** 10.3390/ma16030956

**Published:** 2023-01-19

**Authors:** Sílvia R. Gavinho, Ana Sofia Pádua, Isabel Sá-Nogueira, Jorge C. Silva, João P. Borges, Luis C. Costa, Manuel Pedro F. Graça

**Affiliations:** 1I3N and Physics Department, Aveiro University, Campus Universitário de Santiago, 3810-193 Aveiro, Portugal; 2I3N-CENIMAT and Physics Department, NOVA School of Science and Technology, Campus de Caparica, 2829-516 Caparica, Portugal; 3Associate Laboratory i4HB, Institute for Health and Bioeconomy, NOVA School of Science and Technology, NOVA University Lisbon, 2819-516 Caparica, Portugal; 4UCIBIO—Applied Molecular Biosciences Unit, Department of Life Sciences, NOVA School of Science and Technology, NOVA University Lisbon, 2819-516 Caparica, Portugal; 5I3N-CENIMAT and Materials Science Department, NOVA School of Science and Technology, Campus de Caparica, 2829-516 Caparica, Portugal

**Keywords:** Bioglass^®^, zinc, osteoconduction, antibacterial properties, membranes, PCL, GBR

## Abstract

Polymeric membranes are widely used in guided bone regeneration (GBR), particularly in dentistry. In addition, bioactive glasses can be added to the polymers in order to develop a matrix that is osteoconductive and osteoinductive, increasing cell adhesion and proliferation. The bioactive glasses allow the insertion into its network of therapeutic ions in order to add specific biological properties. The addition of zinc into bioactive glasses can promote antibacterial activity and induce the differentiation and proliferation of the bone cells. In this study, bioactive glasses containing zinc (0.25, 0.5, 1 and 2 mol%) were developed and structurally and biologically characterized. The biological results show that the Zn-containing bioactive glasses do not present significant antibacterial activity, but the addition of zinc at the highest concentration does not compromise the bioactivity and promotes the viability of Saos-2 cells. The cell culture assays in the membranes (PCL, PCL:BG and PCL:BGZn2) showed that zinc addition promotes cell viability and an increase in alkaline phosphatase (ALP) production.

## 1. Introduction

Periodontitis is an inflammatory disease that affects the periodontal tissues and is initiated by the presence of a microbial biofilm, causing gingival bleeding and the degradation of connective tissues and alveolar bone, leading to tooth loss [1,2].

The regeneration of the periodontal region is one of the challenges in this area due to the unsatisfactory results caused by the presence of processes related to inflammation and healing. The treatment used must present a regeneration at the level of all the affected regions, namely the cementum, the periodontal ligament and the alveolar bone [1]. However, epithelial cells, being the first to migrate to the bone defect, may develop near the tooth root, inhibiting bone formation [1,3]. Guided bone regeneration (GBR) has been one of the most effective strategies in clinical practice that allows and promotes osteoblastic proliferation, differentiation and bone regeneration and prevents epithelial cells and gingival tissue from reaching the defect area [3,4,5,6]. Therefore, this technique uses a barrier membrane (BM) to prevent early epithelial migration to the defect site, which gives the periodontal system enough time to regenerate the periodontal ligament, cement and bone [2,4,6]. Moreover, the success of GBR membranes is due to several essential parameters, namely biocompatibility (ensuring integration into the host tissue), cell occlusion, space maintenance and creation, appropriate mechanical and physical properties (flexibility, permeability and degradability), bacteriostatic activity and easy manipulation [6,7,8].

Considering the needs of bone tissue engineering, bioresorbable BMs have been investigated that present improved osteogenic and bioactive properties and also do not require new surgery to remove them, as in the case of non-resorbable membranes [6,9]. One of the most widely used resorbable synthetic polymers in the field of tissue engineering is polycaprolactone (PCL). It is a biocompatible aliphatic polyester with high mechanical strength. However, it has restrictions regarding bone regeneration due to its hydrophobicity, and it is not bioactive. To avoid or minimize this disadvantage, other approaches and adaptations to PCL are required, namely the introduction of bioactive and osteogenic materials, such as bioceramics (hydroxyapatite or tricalcium phosphate) and bioactive glasses [10].

Bioglass^®^, developed by Hench et al., presents one of the best rates of bioactivity compared to other biomaterials, allowing the formation of a layer of hydroxyapatite that presents a similar structural and chemical phase with the mineral composition of bones. This reaction mechanism is due to the ability of rapid ionic exchange between the bioactive glass and the physiological environment in which it is inserted, allowing the migration of Ca^2+^ and PO_4_^3−^ to the surface of the bioactive glass, developing a layer of amorphous calcium phosphate that subsequently transforms into the crystalline layer of hydroxyapatite. In addition, Bioglass^®^ has osteoinductive and osteogenic properties that are due to the fact that it is able to stimulate the recruitment of immature cells and stimulate their differentiation into osteoblasts, increasing their proliferation. Bioglass^®^ also acts as a matrix for cell growth and development; it can promote blood vascularization and, most importantly, it is non-toxic [11,12,13].

In particular, bioactive glass can be presented as a multifunctional material due to its ability to provide more than one biological property with the addition of therapeutic metal ions to its composition. This possibility allows the improvement of the biological activity, mechanical properties, osteoconductivity and also antibacterial activity in the case of the addition of ions, such as Ag, Zn and Ce [14]. In this work, the zinc-containing bioactive glass was studied due to its important role in the proliferation of osteoblast cells while inhibiting bone resorbing osteoclasts in the stimulation of bone formation, in mineralization and also for presenting antibacterial activity [15,16,17,18,19].

In this work, the 45S5 bioactive glass containing 0%, 0.25%, 0.5%, 1% and 2% mol of zinc was characterized at the structural and biological level, such as its cytotoxicity, bioactivity and antibacterial activity. Furthermore, cell adhesion and proliferation and alkaline phosphatase (ALP) activity were evaluated using PCL and composite membranes prepared with solvent casting, hot pressing and salt-leaching.

## 2. Materials and Methods

### 2.1. Glasses Preparation

The bioactive glass composition was synthesized considering the Bioglass^®^ formulation reported by Hench et al. (45 SiO_2_-24.5 Na_2_O-24.5 CaO-6 P_2_O_5_, wt%) [11]. Several percentages of ZnO (0.25, 0.5, 1 and 2 mol%) were added to the Bioglass^®^ (Base). The starting chemicals SiO_2_ (Sigma-Aldrich, Darmstadt, Germany, purity 99.8%), P_2_O_5_ (Sigma-Aldrich, Darmstadt, Germany, purity 99%), CaCO_3_ (Sigma-Aldrich, Darmstadt, Germany, purity ≥99%), Na_2_CO_3_ (Sigma-Aldrich, Darmstadt, Germany, purity ≥99.0%) and ZnO (Sigma-Aldrich, Darmstadt, Germany, purity 99.99%) were mixed and homogenized using a planetary ball-milling process for 1 h at 300 rpm, using agate vessels and balls. The mixed powder was calcined for 8 h at 800 °C. The melt-quenching process was made in a platinum crucible, and the temperature parameters for the melt were 1300 °C for 1 h. The bioactive glass was re-melted under the same parameters to improve the homogeneity of the samples. Using an agate mortar, the bulk material was ground to decrease the particle size and its distribution. After, the powder was milled in a planetary ball mill system for 60 min at 300 rpm.

### 2.2. Structural Characterization

The structure of the bioactive glass samples was characterized with X-ray powder diffraction (XRD) and Fourier-transform infrared spectroscopy (FTIR). The XRD results were obtained on an Aeris-Panalytical diffractometer at room temperature. CuKα radiation (λ = 1.54056 Å) was generated with 40 kV and 15 mA. The scanning parameters were a scan step of 0.002° and a 2θ range of 10° up to 70°. FTIR results were performed with a Perkin-Elmer Spectrum BX FTIR™ spectrometer in the range of 1400–500 cm^−1^ at a resolution of 4 cm^−1^ with 128 co-added scans. During data collection, the room temperature and humidity were kept at approximately 25 °C and 37%, respectively.

### 2.3. Morphological Characterization

The morphology of the sample’s surface was evaluated using a SEM microscope from TESCAN (model Vega 3). A semi-quantitative examination of the chemical composition of the samples was made using the Bruker EDS system coupled to the microscope. Some regions of each sample were analysed using a square scanning area of 100 µm × 100 µm. Prior to the visualization, the sample’s surface was coated with carbon, reducing the surface electron resistivity.

The particle size and distribution were measured by the HORIBA Scientific LA-960V2 wet circulation system. The analysis was based on the principle of laser diffraction, and the particle size calculation was based on the Fraunhofer and Mie models. The measurements were assessed with the samples (Base and Zn2 powder) dispersed in distilled water.

### 2.4. Cytotoxicity Assay

The possible cytotoxic effects of all the samples were assessed according to the “ISO 10993-5 Biological evaluation of medical devices—Part 5: Tests for in vitro cytotoxicity” standard using the extract method and human osteosarcoma cell line (Saos-2 cells, ATCC^®^ HTB-85™). All powders were sterilized at 120 °C for 2 h, and the initial concentration of the extract was 100 mg/mL, being serially diluted to 50 mg/mL, 25 mg/mL and 12.5 mg/mL. For the non-passivated extract, the bioactive glass powder was incubated for 24 h at 37 °C in McCoy’s 5A medium. After incubation, the extract was filtered with a 0.22 µm Millipore filter and stored at 37 °C. In this non-passivated extract, the products collected are related to the first stage of the reaction between the bioactive glass and the cell culture medium. In order to avoid the cytotoxicity related to the first release of ions that causes a pH increase, the samples were passivated. This passivation process protects the cells from the initial burst release of ions and its consequences. For the passivated extract, the same bioactive glass powder was incubated for another 24 h at 37 °C in incubated McCoy’s 5A medium [14,20].

The Saos-2 cell line was seeded in 96-well plates and incubated for 24 h at 37 °C with 5% CO_2_. The culture medium was removed and replaced with passivated and non-passivated extracts. The negative control was viable cells, and the positive control was cells in a cytotoxic environment induced by the addition of 10% dimethyl sulphoxide (DMSO).

After 48 h of cell culture in contact with the extracts, a colorimetric viability assay using resazurin was performed. The solution of resazurin and medium in a *v*/*v* ratio of 1:1 was reacted for 3 h. Using a Biotek ELx 800 UV plate reader, the absorption at the wavelengths of 560 nm and 600 nm was measured [21].

Three biological replicates were performed with six statistical replicates in each to verify the reproducibility of the assay.

### 2.5. Bioactivity

The bioactivity assay was assessed in pellets with a diameter of 7 mm that were pressed for 5 min at 2 tones. Following “Implants for surgery—In vitro evaluation for apatite-forming ability of implant materials” (ISO 23317:2014) and as proposed by Kokubo et al., the bioactivity was analysed [22]. The samples were immersed in simulated body fluid (SBF) and removed from the medium and cleaned with ultrapure water after 12 h, 24 h, 48 h, 96 h, 336 h and 672 h. Similarly, the sample surface was analysed before and at the end of the time intervals with SEM/EDS. To evaluate biodegradation, all the samples were weighed before and after being in contact with SBF. In addition, the pH of the medium in which the pellets were inserted was measured. The assay was performed in duplicate.

### 2.6. Antibacterial Activity

To observe the antimicrobial behaviour of all the samples, the method of agar diffusion assay plates was used with the reference strains *Escherichia coli* K12 DSM498 (DSMZ, Braunschweig, Germany), *Staphylococcus aureus* COL MRSA (methicillin-resistant strain, provided by Rockefeller University) and *Streptococcus mutans* DSM20523 (DSMZ, Braunschweig, Germany). The bacterial strains were cultivated overnight in tryptic soy broth (TSB) at 37 °C. The tested disks with a diameter of 7 mm and ~2 mm of thickness were previously sterilized at 180 °C for 2 h.

The two-layer bioassay was performed using the TSB solidified with agar 1.5% *w*/*v*, base layer, and 0.8% *w*/*v*, top layer. The plates were processed with 18–20 mL base layer and 4 mL of molten seeded overlay containing approximately 10^8^ CFU/mL of the appropriate indicator bacteria. At the centre of the plate, the disks of material to be tested were deposited, and the plates were incubated for 24 h at 37 °C. For *S. mutans*, an incubator kept at 5% CO_2_ was used [14].

Photographs of the pellets were taken, and the diameters of the inhibition halos were measured with ImageJ software. The diameter measurements of each pellet were repeated 50 times in several orientations [23]. The study was performed in three independent assays. The data was statistically analysed with an unpaired *t*-test, comparing the bioactive glass base composition with each of the different samples using GraphPad Prism 8.0 software.

### 2.7. Preparation of PCL and BG/PCL Scaffolds

The polycaprolactone (PCL)/ Bioglass^®^ composite scaffolds were prepared with solvent casting, hot pressing and salt-leaching. PCL (Sigma Aldrich) was dissolved at a fixed concentration of 20 wt% in acetone, and 10 wt% salt (NaCl) particles (100–200 µm) were added to the solution. The composite solutions also included 5 wt% of Base (BG) or 5 wt% of Zn2 (BGZn2).

The solutions were placed in a magnetic stirrer with vigorous stirring overnight to form a homogeneous slurry system. These solutions were cast to produce films from which 20 mm diameter disks were cut.

The polymeric and composite were crafted using a hot platen press at 60 °C and a pressure of 4 tons. The salt was then leached out by immersing the samples in distilled water with constant agitation for a 24 h period and then dried.

### 2.8. Cell Culture

#### 2.8.1. Adhesion and Proliferation

The ability of the scaffolds to support cell metabolism was evaluated through cell adhesion and proliferation studies.

The scaffolds were sterilized with 70% ethanol and washed with PBS. Then, the materials for the cell culture and material controls were placed in 24-well plates and fixed with silicone O-rings.

Saos-2 cells were seeded at a concentration of 30 k cells/cm^2^ directly over the sample’s surface and at the bottom of the wells for the cell controls. The cells were maintained in McCoy’s 5A medium and incubated for 24 h at 37 °C in a controlled 5% CO_2_ atmosphere.

The cell adhesion rate was determined by the reduction in resazurin, as in the cytotoxicity tests. For this process, the medium was substituted by a 1:1 solution of resazurin/McCoy’s medium and incubated for 3 h. Then, the incubated media was transferred to a 96-well plate, and the absorbance at 570 nm and 600 nm was read in a microplate reader (Biotek ELx 800 UV) [24]. The resazurin assay was repeated at 3 and 7 days to evaluate the cell proliferation in each of the materials.

The data of the three independent biological assays were statistically analyzed using an ANOVA, comparing all samples in each time point and the same samples throughout the time. The statistical analysis was performed using OriginPro software with 95%, 99% and 99.5% levels of significance.

#### 2.8.2. Alkaline Phosphatase (ALP) Activity

ALP is an enzyme expressed by cells during osteogenesis and is well established as a differentiation marker. A colorimetric assay was used to measure the ALP expression. This reaction used 8 mg/mL of 4-nitrophenyl phosphate disodium salt (Sigma-Aldrich) dissolved in tris-hydrochloric acid solution.

This essay consisted of filtering the medium that was in contact with the samples and reading the absorbance at 405 nm to obtain the baseline. Then, the ALP solution was added in a 1:1 ratio to the medium. This solution was incubated for 30 min, and the absorbance was measured.

The statistical analysis performed to check the significance between each sample and their growth during the experience was the same as the one described for the adhesion and proliferation assay.

## 3. Results and Discussion

### 3.1. Structural and Morphological Characterization

The structural characterization of all the synthesized powders was studied with XRD and FTIR spectroscopy. The XRD patterns demonstrated by the single broad band between 25–40° 2θ for all compositions are typical of the amorphous silicate phase as shown in Figure 1 [17,25,26]. The results do not show peaks related to Zn^2+^-based crystals (e.g., ZnO), which reveal the incorporation of Zn^2+^ into the glass network as a modifier without changing the matrix structure [18]. It is crucial to maintain the structure of the glass matrix without compromising the bioactivity rate because the dissolution and ion release kinetics depend on the structure of the glass network and on the type of ions present in the glass [27].

The FTIR spectra of the bioactive glasses with several concentrations of zinc ion are presented in Figure 2. The typical absorption bands of the amorphous Bioglass^®^ can be observed for all samples. These results are in agreement with the XRD patterns and do not show structural modification, comparing the base Bioglass^®^ with the bioactive glass containing zinc up to 2 mol%. The bands around 1029 cm^−1^ and 929 cm^−1^ are related to the Si-O-Si asymmetric stretching mode; the bands around 732 cm^−1^ and 485 cm^−1^ are associated to the Si-O-Si symmetric stretching mode and Si-O-Si bending mode, respectively. The shoulder at 597 cm^−1^ is related to the P-O bending mode from amorphous phosphate observed for the Bioglass^®^ [28,29,30,31,32,33].

Figure 3 shows the size distribution of the (a) base Bioglass^®^ and (b) Zn2 samples after the previously described grinding process. The average size of the particles or agglomerates of the base Bioglass^®^ particles is approximately 5.2 µm. Considering the standard deviation, the sample with 2 mol% of ZnO presents average size values in the same size range (4.5 µm). However, it presents a slight increase in size distribution and presents particles with smaller sizes below 1 µm.

### 3.2. Cytotoxicity Assay

Figure 4 shows the results of the Saos-2 cell line viability in contact with the bioactive glass extracts (non-passivated and passivated). The effect of the extracts on the growth of the Saos-2 cell population was measured by the resazurin assay. This evaluation allows us to determine if the compositions are harmful to the organism and can be used in the bone regeneration field. The results show that the non-passivated extracts with McCoy’s culture medium present a severe level of cytotoxicity at the concentration of 100 mg/mL. For the concentration of 50 mg/mL, the same cytotoxic behaviour is evident for all samples except for sample Zn2, which shows no cytotoxicity [17,34,35,36]. For the concentration of 25 mg/mL, all the extracts present cell viability superior to 80%, being considered non-cytotoxic, with the exception of the non-passivated extract of the Base Bioglass^®^ that remains moderately cytotoxic. Furthermore, the results at 25 mg/mL suggest that the passivation of the samples allows a decrease in the effects caused by the alkalinization of the pH induced by the bioactive glasses. These data allow us to verify that, at this concentration, the samples are no longer toxic in our organism due to the natural process of pH regulation that occurs in vivo [20,37]. This effect is verified for the higher concentrations of the extracts (50 mg/mL) for the passivated extracts of the zinc-containing bioactive glasses that do not show any cytotoxicity towards Saos-2 cells. All the samples (passivated and non-passivated extracts) present non-cytotoxicity for concentrations of extract at 12.5 mg/mL.

### 3.3. Bioactivity

The evaluation of the chemical dissolution process of the bioactive glasses in general is crucial because it is related to the bioactivity and to the capability to form a new bone when inserted in the body. The bioactive glass pellets were immersed in SBF for 12 h, 24 h, 48 h, 96 h, 336 h and 672 h according to the ISO standard (ISO 23317:2014 Implants for surgery—In vitro evaluation for the apatite-forming ability of implant materials). The sample’s surface was evaluated with SEM-EDS. The quantification in atomic % of each ion (Si, Na, P and Ca) on the sample’s surface for all times of SBF immersion was performed as shown in Figure 5. The evolution of these ions with time allows us to observe the deposition of the apatite layer on the sample surface. The reaction mechanism of the bioactive glass immersed in SBF involves a faster glass–fluid ion exchange between Na^+^ and Ca^2+^ from the bioactive glass with H^+^ from SBF. This process promotes the formation of Si-OH and, consequently, the rise of pH. The alkalinization of the medium promotes the formation of soluble silica by the attack of Si-O-Si. This silica gel layer can absorb ions from the SBF medium and allows the diffusion of Ca^2+^ and PO_4_^3−^ ions through the silica gel, leading to the formation of an amorphous calcium–phosphate layer at the bioactive glass surface following its crystallization and the formation of hydroxyapatite [11,12,38,39,40]. Figure 5 shows the atomic % for Si, Na, P, Ca and Ca/P on the surface of the pellets for all times of SBF immersion. The graph for Si ion (Figure 5a) shows a decrease in its percentage over time until it approaches zero at 96 h; this phenomenon is associated with its dissolution in the medium and the deposition of the Ca/P layer on the bioactive glass. Its dissolution has, as a consequence, the release of Na ions into the medium as observed in Figure 5b. The formation of the layer is also confirmed with the increase in Ca and P ions on the surface over time as shown in Figure 5c,d that starts to reach stability after 96 h. The extent of the Ca/P rich layer is demonstrated in Figure 5e, which indicates an approximation of the Ca/P ratio value of ≈1.67 present in the hydroxyapatite for all samples with the samples Zn2 being the closest to this value (Ca/P ≈ 1.66) from 48 h [41,42,43]. Broadly, the evaluation of the atomic percentage of the ions on the sample surface showed that the addition of the zinc ion did not modify the structure of the Base Bioglass^®^ or its dissolution rate in a way that could influence the bioactivity of the glass.

The morphology of the Base and Zn2 samples’ surface after immersion in SBF for 24 h, 96 h and 672 h was observed with SEM as shown in Figure 6. The images reveal that there is a deposition of particles with spherical morphology on the surface of the samples from 24 h, increasing in quantity and size over the time of immersion in SBF. The Zn2 sample shows no significant differences compared to the base in agreement with the EDS results. After 24 h of immersion, the base and Zn2 samples present particle size diameters between 200–400 nm and 300–600 nm, respectively. After 96 h, the presence of agglomerates is remarkable in both samples, and the filled layer rich in Ca/P is visible in the base and the Zn2. The base Bioglass^®^ presents agglomerates of particles with diameters of 2–3 µm, and on the surface of the Zn2 sample, we observe smaller agglomerates of particles of the order of 1 µm. At 672 h, this layer became even more evident, and the particle agglomerates became larger in both samples [44]. Thus, this characterization confirms that zinc does not negatively influence the formation of the Ca/P-rich layer, and these compositions can be used as osteoconductive materials in bone regeneration.

Figure 7 shows the evolution of the pH of the media at the various immersion times, presenting a set of measurements for the samples in which the media was replaced every two days (red rectangle) and for samples in which the media was not replaced (blue rectangle). According to the reaction mechanism of the bioactive glass, when immersed in SBF, it is verified that the pH shows an evident increase in the first 48 h of immersion reaching a plateau after 96 h with a pH close to 9 for the samples that did not have a change in the medium. The replacement of the medium every two days allowed for mimicking, as best as possible, the presence of a continuous flow of biological fluids when the Bioglass^®^ is placed in the organism, allowing to verify the decrease in the pH along the immersion time. From 96 h, a stabilization between 7.8 and 8 is observed, which is related to the formation of the apatite layer on the surface of the bioactive glass pellets [45].

### 3.4. Antibacterial Activity

Figure 8 shows the measurements of the inhibition halo diameters for all samples against *E. coli*, *S. aureus* and *S. mutans*. None of the Zn-containing bioactive glass compositions tested showed significant antibacterial activity against the microorganisms used. The basis behind this method is the direct correlation between the inhibition of bacterial growth and the dissolution of antimicrobial components from the disk. The antibacterial effect of ZnO is widely known; however, the literature also reports an absent or minor antibacterial activity using this methodology, which might be related to the low release of the ion during the assay. Therefore, a longer diffusion time is suggested in order to allow the release of a greater amount of ions [34,46].

### 3.5. Membranes Characterization

The morphology of the membranes (PCL; PCL:BG and PCL:BGZn2) was evaluated with SEM as shown in Figure 9. The micrographs show that the addition of bioactive glass to the composite did not influence the microstructure and porosity. All the membranes present macro and micro porosity confirming that the pore diameters present values of 100–200 nm, resulting from the NaCl that was used as the template. Moreover, the pores present interconnectivity, presenting conditions for cell adhesion and proliferation [47,48,49].

Figure 10 shows the percentage of cell adhesion and proliferation for 1, 3 and 7 days on the membranes. The results reveal that there is cell adhesion in all samples, but the presence of the bioactive glass shows significantly lower adhesion percentages than the PCL membrane. Regarding the cell proliferation rate, the PCL membrane and composites with BG and BGZn2 show an increase in cell population, indicating cell proliferation over time. The highest proliferation rate is verified in the PCL:BGZn2 membrane compared to the PCL and PCL:BG as shown in Table 1.

Analysing ALP production (Figure 11), the PCL:BGZn2 sample shows higher ALP production, similar to that produced by the cell control. This indicates, as in the cytotoxicity results (Figure 4), that zinc ions promote cell viability and an increase in ALP activity [50,51,52]. In the case of PCL:BG, there is hindrance of ALP production, which may be related to the low proliferation rate of the PCL:BG composite between days 3 and 7. This decrease can be associated with the cytotoxicity of the powder without passivation in agreement with the results of cellular viability. As mentioned above, in order to improve the biocompatibility of the composites, the powders should be subjected to prior passivation to increase the viability and, consequently, proliferation.

## 4. Conclusions

All samples containing zinc synthesized with melt-quenching showed an amorphous structure identical to Bioglass^®^. The cytotoxicity assay shows that the passivation process increases Saos-2 cell viability for all samples with the sample containing the highest zinc concentration showing cell viability above 80% at 50 mg/mL without passivation. The bioactivity results showed that the addition of zinc did not compromise the ability to form the apatite layer on the surface of the sample. However, the results of the antimicrobial activity showed that the methodology used did not allow the release of the ions and, consequently, an inhibitory effect. Thus, the results suggest an increase in the diffusion time of the samples in future studies. The results of the cell adhesion and proliferation in the membranes suggest, as a future adjustment in the process, a previous passivation in order to allow a higher proliferation rate in the composites with bioactive glass. The cell proliferation, cell adhesion and ALP production results also suggest that the most promising membrane for application in tissue regeneration is the composite containing Zn2 (PCL:BGZn2).

## Figures and Tables

**Figure 1 materials-16-00956-f001:**
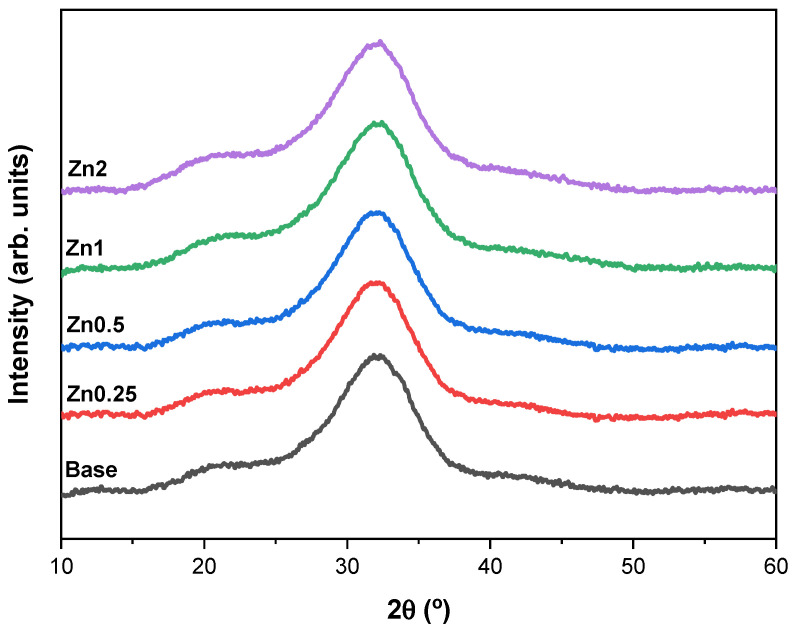
XRD diffractogram of the base Bioglass^®^ and the bioactive glasses with different concentrations of zinc (0.25, 0.5, 1 and 2 wt%).

**Figure 2 materials-16-00956-f002:**
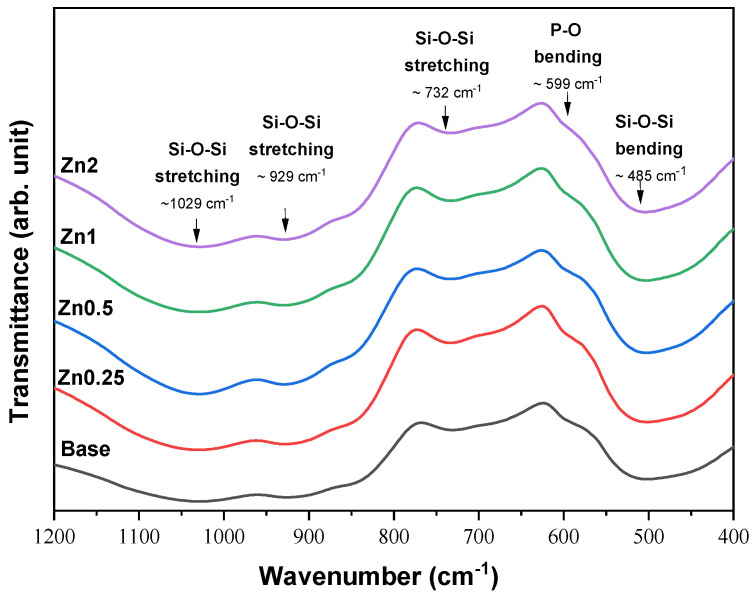
FTIR results of the base Bioglass^®^ and the bioactive glasses with several concentrations of zinc (0.25, 0.5, 1 and 2 wt%).

**Figure 3 materials-16-00956-f003:**
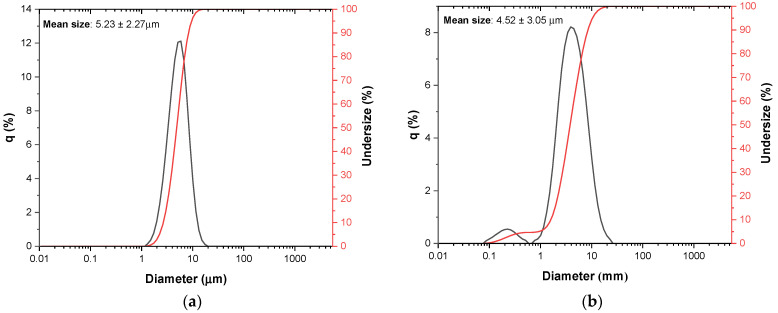
Particle size distribution with laser granulometry for (**a**) base Bioglass^®^ and (**b**) Zn2.

**Figure 4 materials-16-00956-f004:**
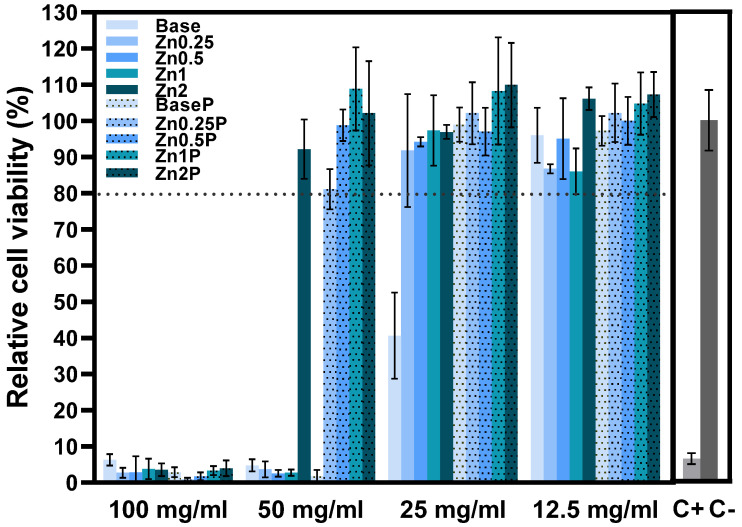
Relative cell viability of Saos-2 cells after 48 h incubation with non-passivated (Base, Zn0.25, Zn0.5, Zn1 and Zn2) and passivated (BaseP, Zn0.25P, Zn0.5P, Zn1P and Zn2P) bioactive glass extracts. Dot line is related to the percentage of cell viability above which extracts are not considered cytotoxic.

**Figure 5 materials-16-00956-f005:**
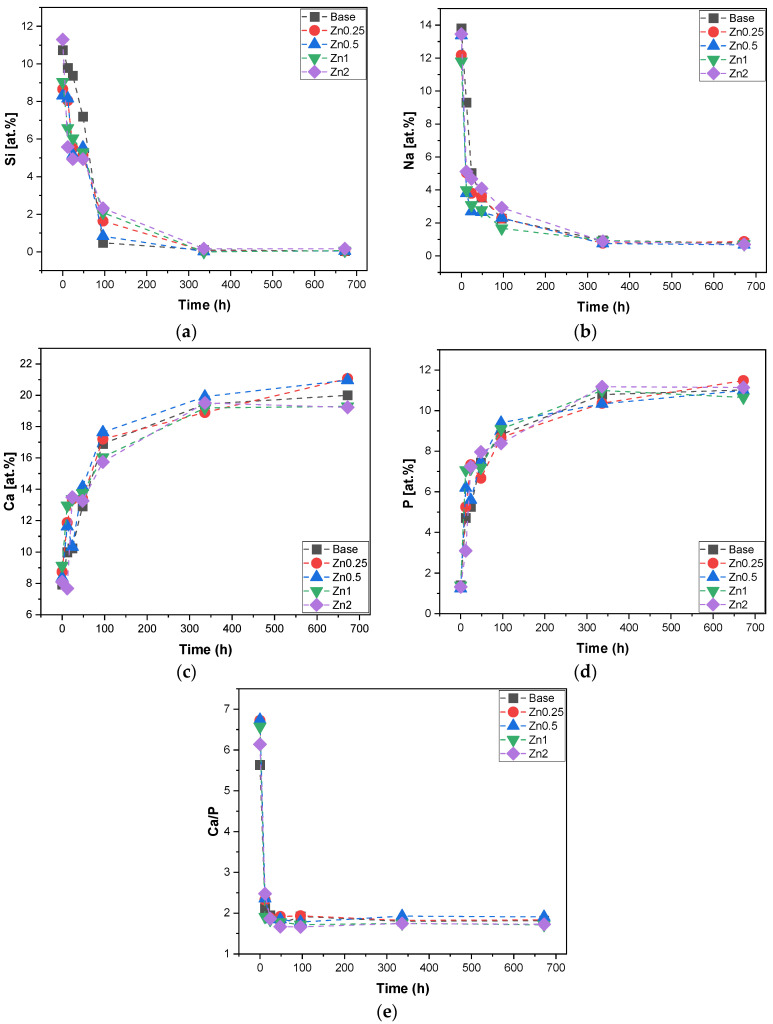
EDS results of the pellet’s surface after SBF immersion for 12 h, 24 h, 48 h, 96 h, 336 h and 672 h. (**a**) Si at.%; (**b**) Na at.%; (**c**) Ca at.%; (**d**) P at.% and (**e**) ratio between Ca at.% and P at.%.

**Figure 6 materials-16-00956-f006:**
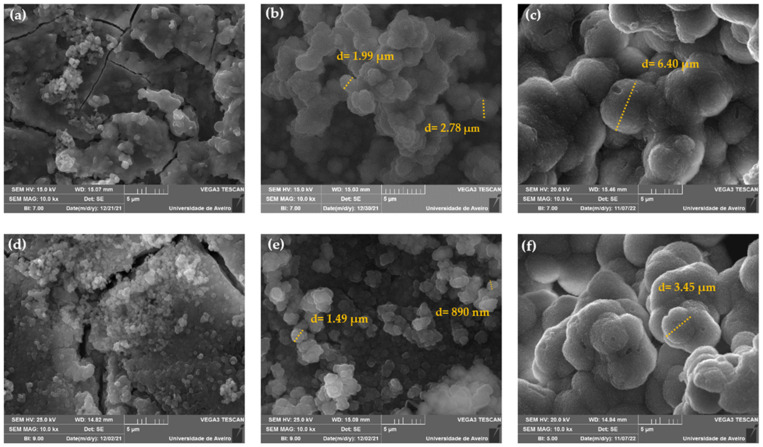
SEM micrographs taken with a magnification of 10 kX of the bioactive glass pellets after 24 h, 96 h and 672 h of immersion in SBF: (**a**) Base 24 h, (**b**) Base 96 h, (**c**) Base 672 h, (**d**) Zn2 24 h, (**e**) Zn2 96 h and (**f**) Zn2 672 h.

**Figure 7 materials-16-00956-f007:**
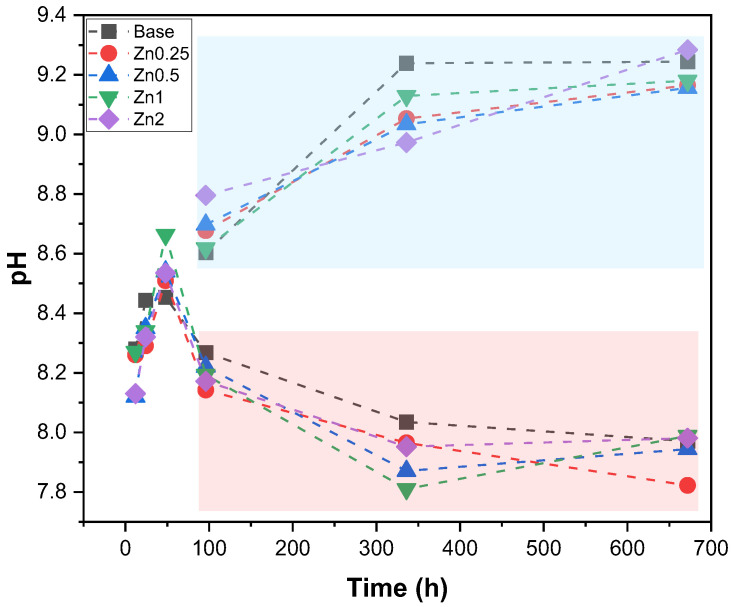
pH values of SBF medium after immersion of all samples for 12 h, 24 h, 48 h, 96 h, 336 h and 672 h. Red rectangle: medium change every two days; Blue rectangle: without medium change.

**Figure 8 materials-16-00956-f008:**
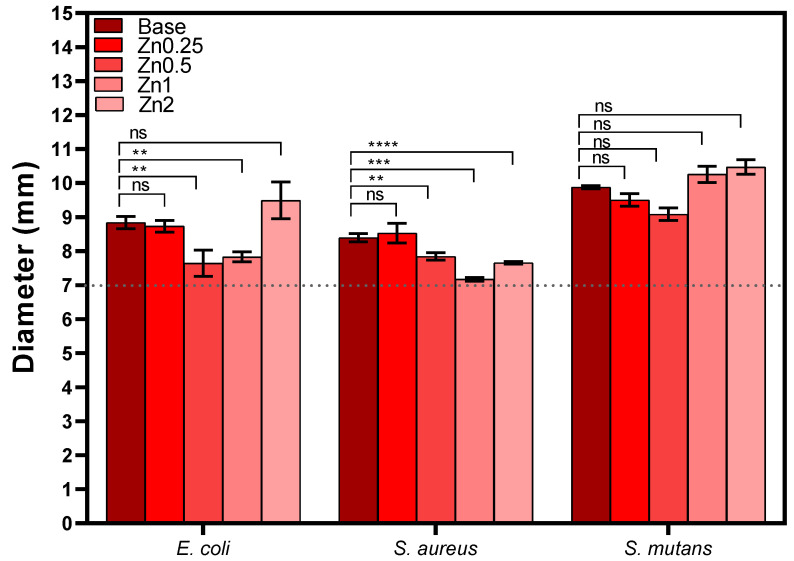
Values of the inhibition halo diameters of all samples against Gram-negative (*E. coli*) and Gram-positive (*S. aureus* and *S. mutans*) bacteria after incubation for 24 h (ns: non significant; ** *p* ≤ 0.01; *** *p* ≤ 0.001; **** *p* ≤ 0.0001).

**Figure 9 materials-16-00956-f009:**
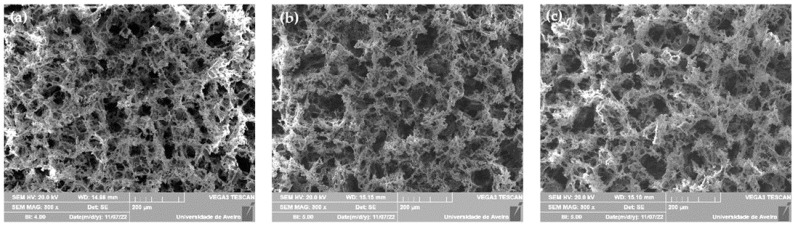
SEM micrographs (magnification: 300×) of the membranes (**a**) PCL, (**b**) PCL:BG and (**c**) PCL:BGZn2.

**Figure 10 materials-16-00956-f010:**
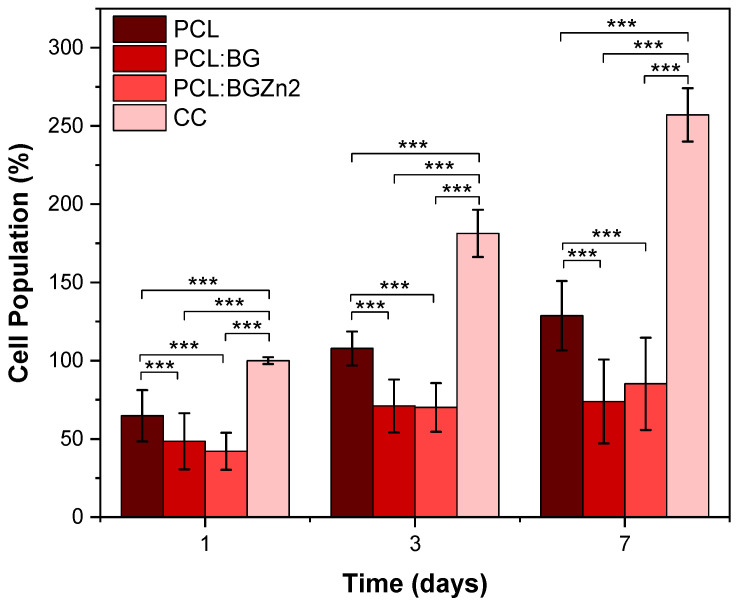
Saos-2 adhesion and proliferation rate for 1, 3 and 7 days in PCL, PCL:BG and PCL:BGZn2 membranes (*** *p* < 0.005).

**Figure 11 materials-16-00956-f011:**
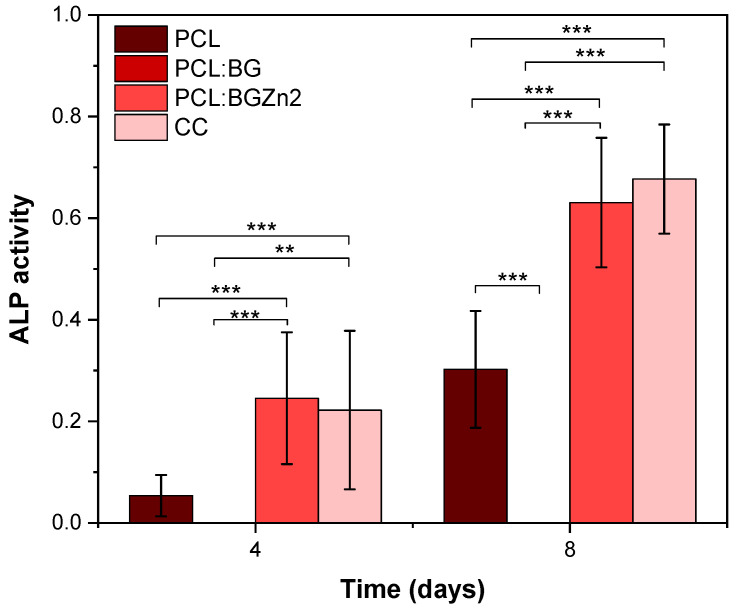
ALP production on days 4 and 8 for PCL, PCL:BG and PCL:BGZn2 membranes (** *p* < 0.01 and *** *p* < 0.005).

**Table 1 materials-16-00956-t001:** Cell proliferation evaluation for the composite membranes with the ratio between cell population on day 7 and on day 1.

Samples	Cell Proliferation
PCL	1.99 ± 0.19
PCL:BG	1.52 ± 0.22
PCL:BGZn2	2.02 ± 0.21
CC	2.57 ± 0.10

## Data Availability

Not applicable.

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
