# Peer review of "Fabrication, Structural and Biological Characterization of Zinc-Containing Bioactive Glasses and Their Use in Membranes for Guided Bone Regeneration"

_materials, 2023, doi:10.3390/ma16030956_

Round 1

Reviewer 1 Report

This is an interesting paper reported novel results from well-conducted experiments.

Some minor improvements are suggested:

1- This paper on periodontal biomaterials deserves to be cited:

Biomaterials, current strategies, and novel nano-technological approaches for periodontal regeneration. Journal of Functional Biomaterials 2019;10:3

2- Glass particle size distribution and related parameters should be reported.

3- Figure 3: statistical analysis is apparently missing (p-value).

4- Mechanical properties were not investigated: have you no available result to add? This is an important point.

5- Please carefully check the reference list so that all the bibliographic info provided for each paper are complete.

Author Response

File

Reviewer 2 Report

Review comments: 

1. There are extensive descriptions of polymer membranes in the abstract and introduction, but the titles, experimental procedures and even the results do not reflect the corresponding effort.

2. The abstract is more like a summary of the introduction than a rough summary of the article.

3. There is a description of BG in the introduction, which favors the addition of therapeutic metal ions, but no description of the effect of BG on bone formation.

4. The description of the preparation of the BG/PCL scaffold is somewhat confused; the name of the scaffold is inconsistent.

5. In antibacterial activity, the bacteriostatic zone method requires the sample to be well-diffusible in agar. This can be done with the oscillatory method, and results will be better if its diffusivity is not good enough.

6. In the cytotoxicity assay, the explanation of “passivated” treatment should be added and explain what changes have been made before and after treatment.

7. Some prospects for follow-up experiments can be mentioned in the conclusions.

8. It is better to number the subheadings in the characterization test section and label the results and discussion separately.

9. Besides, there are some mistakes in writing. Please correct them carefully. For example, line 145 should be 10 to the eighth power; Line 211, should state is "non-passivated" sample; Figure 4 is not very clear, you can do it differently; Are the labels in Figure 5 wrong? Line 276 should read Figure 6.

Author Response

File

Round 2

Reviewer 2 Report

English language and style are fine/minor spell check required.